# Charge Displacement Analysis—A Tool to Theoretically Characterize the Charge Transfer Contribution of Halogen Bonds

**DOI:** 10.3390/molecules25020300

**Published:** 2020-01-11

**Authors:** Gianluca Ciancaleoni, Francesca Nunzi, Leonardo Belpassi

**Affiliations:** 1Dipartimento di Chimica e Chimica Industriale, Università degli Studi di Pisa, via Giuseppe Moruzzi 13, 56124 Pisa, Italy; 2Dipartimento di Chimica, Biologia e Biotecnologie, Università degli Studi di Perugia, via Elce di Sotto 8, I-06123 Perugia, Italy; francesca.nunzi@unipg.it; 3Istituto di Scienze e Tecnologie Chimiche “Giulio Natta” del CNR (SCITEC-CNR), via Elce di Sotto 8, I-06123 Perugia, Italy; leonardo.belpassi@cnr.it

**Keywords:** halogen bond, bond analysis, non-covalent interactions

## Abstract

Theoretical bonding analysis is of prime importance for the deep understanding of the various chemical interactions, covalent or not. Among the various methods that have been developed in the last decades, the analysis of the Charge Displacement function (CD) demonstrated to be useful to reveal the charge transfer effects in many contexts, from weak hydrogen bonds, to the characterization of σ hole interactions, as halogen, chalcogen and pnictogen bonding or even in the decomposition of the metal-ligand bond. Quite often, the CD analysis has also been coupled with experimental techniques, in order to give a complete description of the system under study. In this review, we focus on the use of CD analysis on halogen bonded systems, describing the most relevant literature examples about gas phase and condensed phase systems. Chemical insights will be drawn about the nature of halogen bond, its cooperativity and its influence on metal-ligand bond components.

## 1. Introduction

Some years ago, Frenking and Krapp compared chemical bonds to unicorns [1], never-seen entities of which everybody talks. Actually, experimental techniques developed so much that in the last years it has been possible to obtain pictures of single molecules, as Atomic Force Microscopy [2,3,4,5]. But even if we are approaching to an experimental characterization of chemical bonds, theoretical bond analysis remains central to deeply understand the nature of chemical bonds in all its variations.

In their contribution, Frenking and Krapp explain the potential of Energy Decomposition Analysis (EDA) [6] to reliably characterize both covalent and “non-covalent” interactions, underlining not only the differences that exist between them but also and, more importantly, their similarities.

And, indeed, EDA is often the basis for any bond analysis but many other tools have been developed in the last decades, as the QTAIM [7], other energy decomposition schemes based on Symmetry Adapted Perturbation Theory (SAPT) [8] or based on block-localized wavefunction theory (BLW) [9], a partitioning of the orbital contribution of the bond into chemically relevant channels [10,11] and within the Local Pair Natural Orbital Coupled Cluster Framework [12].

Each of them can be used to obtain information and each of them works differently, with pros and cons.

Some years ago, the Charge Displacement (CD) function was introduced to study the chemical bond between noble metals and noble gases [13] and then applied to various kind of chemical interactions. These include the quantitative study of charge transfer in hydrogen bonding [14,15,16] and halogen bonding [17,18] in weakly bound adducts, to characterize charge transfer (CT) effects on conduction band of dye sensitized solar cells [19] and also to describe the charge rearrangement in the electronic excited states [20]. Above all, it provided to be extremely versatile in coordination chemistry. Indeed, thanks to the possibility of decomposing the charge rearrangement upon the bond formation between two species into contributions (see Methodological Aspects), it has been extensively used to characterize metal-ligand bonds [21,22,23,24,25] providing, in some cases, an elegant theoretical framework to better rationalize experimental results [26,27,28,29]. Very recently, CD analysis was the key tool for revealing unexpected acceptor properties of helium atoms [30,31]. For weak interactions, such as chalcogen [32,33] or hydrogen bonds [14], the fragmentation is particularly straightforward and generally there is no need to decompose the difference in electronic density between the adducts and the fragments (Δ*ρ*) into contributions, as only the total is relevant. Some exceptions have been reported, for example in the case of bifurcated halogen [34] and chalcogen [35] bonding or with urea [36] and selenourea [37]. On the other hand, it must be said that the CD provides information about the orbital part of the interaction (charge transfer and polarization), which may or may not be the most relevant one, especially in the case of weak interactions. For this, CD has been often associated to an Energy Decomposition Analysis, in order to verify the importance of the orbital part with respect to the other contributions (electrostatics, Pauli repulsion, dispersion, etc.).

This is particularly important for halogen bonding (XB), which has been defined as the “attractive, non-covalent interaction that can form between an electrophilic region of a halogen atom in a molecule and a nucleophilic region of a molecule [38].” The presence of the word “non-covalent” in the definition seems to indicate that the XB is essentially electrostatic in nature, as some papers claim [39,40,41]. But many other authors do not agree with this and clearly recognize the importance of the orbital contribution, as theoretical [42] and experimental [43,44] studies underline. An interesting review titled “Halogen Bonding: A Halogen/Centered Non-covalent interaction yet to be understood” by Varadwaj et al. [45] reports the conflicting views in the field. As an example of the importance of the orbital contribution to the halogen bond, we mention a recent contribution by Rosokha and coworkers [43] that co-crystallized a large number of XB adducts using 1,4-diazabicyclo[2.2.2]octane (DABCO) and bromine electrophiles. The distance between the nitrogen of the DABCO and the bromine ranges from 2.130 (DABCO^…^Br_2_) to 2.910 (DABCO^…^1,4-dibromotetrafluoro-benzene) Å, demonstrating that there is no discontinuity passing from a bond that is generally accepted as “covalent” to a weaker interaction commonly indicated as “non-covalent” and the two extreme cases are more similar in nature than generally thought. Similar conclusions have been drawn by Eraković and others, studying the crystallographic charge density of strong N^…^Br XBs [46].

In this review, we will show how the CD function analysis can be useful in the theoretical characterization of XBs, especially in a combined experimental-theoretical strategy. Examples for both small and complex systems will be presented, in order to better illustrate how an in-depth knowledge of the electronic fluxes can be useful to have information on the physical nature of XBs.

## 2. Methodological Aspects

The Charge Displacement analysis is based on Equation (1) [22]. Δ*ρ*(*x*,*y*,*z*) is the difference between the electron density of a complex and that of its non-interacting fragments placed in the same position as they occupy in the complex. In the present case, the fragmentation depends on the interaction under examination and are generally indicated in each case. The function Δ*q*(z) defines, at each point z’ along a chosen axis, the amount of electron charge that, upon formation of the bond between the fragments, moves across a plane perpendicular to the axis through the point z.’ A positive (negative) value corresponds to electrons flowing in the direction of decreasing (increasing) z. Charge accumulates where the slope of Δ*q* is positive and decreases where it is negative.
(1)$$\Delta q(z’)\;=\;\int_{-\infty }^{+\infty }dx\int_{-\infty }^{+\infty }dy\int_{-\infty }^{z’}dz\Delta \rho \left(x,y,z\right),$$
where ∆*ρ*(*x*,*y*,*z*) is the difference between the electron densities of a complex and the sum of that of its non-interacting fragments, frozen at the same geometry they assume in the complex. Because the key quantity is the electron density, the CD method can be used in combination with any theoretical quantum chemical method. This includes approximated single determinant like wavefunction methods such as the Hartree/Fock (or Kohn-Sham method) or even explicit highly correlated methods such as coupled cluster, multi-reference configuration interaction or even the full configuration interaction. Noteworthy, the opportunity to use highly correlated methods was crucial to reveal small CT components in weakly interacting systems containing hydrogen and halogen bond with noble gases [14,31].

In those cases in which the molecular system has a symmetry and the electron densities (of both adduct and fragments) are worked out using a single determinant method as Hartree-Fock (or more commonly Kohn/Sham theory), the electron density difference, ∆*ρ*, can be separated in contributions following the irreducible representations of symmetry point group to which the molecule and fragments belong [22].

For non-symmetric systems, we make use of the natural orbital for chemical valence theory (NOCV) [10,47]: Δ*ρ*’ is built from the occupied orbitals of A and B, suitably orthogonalized to each other and renormalized (*promolecule*), using the “valence operator” (Equation (2)) [48,49,50],
(2)$$\hat{V}\;=\;\sum_{i}\left(\left|\psi_{i}^{\left(AB\right)}\rangle \langle \psi_{i}^{\left(AB\right)}|-|\psi_{i}^{0}\rangle \langle \psi_{i}^{0}\right|\right),$$
where *ψ_i_*^0^ is the set of the occupied Kohn−Sham orbitals of fragments A and B, mutually orthonormalized and *ψ*_i_^(*AB*)^ is the set of occupied orbitals of the adduct. Despite the similar notations, Δ*ρ*’ and Δ*ρ* are computed in different ways and should not be confused. The NOCVs can be grouped in pairs of “complementary” orbitals (*φ_k_*, *φ_−k_*) corresponding to eigenvalues with same absolute value but opposite sign (Equation (3)).
(3)$$\hat{V}\varphi_{\pm k}\;=\;\pm \nu_{k}\varphi_{\pm k\;}(\nu_{k}>0),$$
where *k* numbers the NOCV pairs (*k* = 0 for the largest value of |*ν_k_*|). In this framework, Δ*ρ*’ can be defined as in Equation (4).
(4)$$\Delta \rho^{\prime }\;=\;\sum_{k}\nu_{k}\left({\left|\varphi_{k}\right|}^{2}-{\left|\varphi_{-k}\right|}^{2}\right)\;=\;\sum_{k}\Delta \rho {’}_{k}.$$

For each value of *k*, an energy contribution associated with the *k*-th NOCV pair is given.

Now the different Δ*ρ*’*_k_* can be separately integrated using Equation (2). This approach has been referred in literature as Charge Displacement via the Natural Orbital for the Chemical Valence method (NOCV-CD) [51].

Before passing to describe those cases in which the CD analysis has been applied in the context of the halogen bond, it is useful to give an illustrative/didactical example of its application to a simple system.

For this purpose, we describe the use of CD analysis in the noble gas/noble metal interaction in the linear molecule Xe-AuF, which is exactly the same system by which the CD analysis was introduced so far [12]. The nature of the chemical bond between two atoms that were typically considered prototypical of chemical inertia and attracted great attention in the early 2000s when Gerry et al. were able to characterize the NgAuX complexes, with Ng = Ar, Kr, Xe; X = F, Cl, Br in gas phase [52,53]. These systems, detected and analyzed using rotational spectroscopy, are linear and present a short Ng−Au bond distance, suggesting that in these complexes the noble gas/noble metal bond had a certain covalent character. However, the issue was questioned and remained controversial for many years.

As insightful approach to understanding the nature of the Ng−Au interaction, we decided to look at a graphical representation of the changes in the electron density upon formation of the chemical bond between the fragments. In Figure 1 (upper panel) we report the contour plots of electron density difference between the complexes and the non-interacting fragments, Xe and AuF (in the same positions they occupy in the adduct) for the XeAuF. The red contours refer to positive differences (density accumulation) and the blue contours are negative ones (density depletion). The dashed contour marks the isodensity value of both isolated fragments which crosses the internuclear axis at the same point and may thus serve to visualize tangent boundaries enclosing the non-interacting fragments.

Concerning the Xe−Au interaction, the most significant feature emerging from the density difference plots is the evident accumulation of electronic charge in the region between the Xe and Au nuclei. Note that this density increase appears to be centered surprisingly close to the point of contact of the tangent fragment boundaries chosen above. However, the simple visual inspection of the electron density deformation could not give a quantitative picture and thus, answer whether there was a covalent contribution in this interaction or not. Introducing the CD function, we were able to quantify the number of electrons that move from a fragment to the other upon the bond interaction and thus gave us a clear picture of the interaction. The meaning of the CD function is very simple.

Imagining each point on the internuclear axis to identify a perpendicular plane passing through that point, the corresponding value of Δ*q* measures the amount of electronic charge that, with respect to the situation in the non-interacting fragments, has moved from the right side to the left side of the plane. Thus, a negative value indicates a charge transfer of that magnitude from left to right. The difference between two Δ*q* values gives the net electron influx into the region delimited by the corresponding two planes. Thus, the regions of the Δ*q* curve where the slope is negative correspond clearly to zones of charge depletion (black contours), while charge accumulates where Δ*q* picks up (red contours).

The first thing one notices is that Δ*q* is negative everywhere in the molecular region, indicating that, at each point, there is a net shift of charge toward the fluorine end upon the formation of the adduct. This is indeed true even to the right of the fluorine position, in accord with the polarization of fluorine observed earlier. The plot makes clear the pronounced electron depletion around the Ng site and the charge transfer to the Ng−Au internuclear region. The pattern of charge transfer is:  the electron loss takes place until about 1.8 Å from the gold site, where there is a rather sharp inversion and charge starts to re-accumulate rapidly until about 0.8 Å, when most of the lost charge (to within less than 0.04 electrons) is recovered. To the extent that the isodensity contour chosen above as delimiting the Ng and AuF fragments has any realistic meaning, we interpret |Δ*q*| at the point of contact (vertical dashed line) as an estimate of the charge transferred two fragment, Ng to AuF in this case (about 0.1 e). In the following we will refer this point of contact as “isodensity boundary.” The picture that emerged for the Xe/Au bond was clear and for this interaction we unquestionably demonstrated that a such unconventional interaction has a significant stabilization from the charge transfer component.

Finally, it should be underlined that CD results are very stable with respect to the exchange–correlation functional, the basis set and the level of theory to account for the relativistic effects (scalar or full four-component Hamiltonian) [29].

## 3. Applications

### 3.1. Gas Phase

The nature, strength and selectivity of halogen bonds in prototype systems have been explored by applying an integrated theoretical-phenomenological approach, based on high level ab-initio calculations and gas phase molecular beam scattering experiments [17,18,54]. Starting from the experimental cross section data, which gives precious information on the interaction potential at both short and long range and applying a semi-empirical iterative procedure, we ended up with an analytical representation of the interaction potential energy surfaces (PESs) based on few physically meaningful parameters [54]. The electrostatic component, usually dominant in the intermolecular halogen bond interactions, is expected small or even vanishing in exemplary systems with high orientational symmetry, such as the complexes of noble gases with di-halogen molecules, Ng-X_2_. In such peculiar cases, the non-electrostatic components, related to Pauli size repulsion, dispersion/induction attraction and charge transfer, have the chance to emerge and can be properly identified.

The experimentally derived PESs are further validated by a comparison with the results from accurate ab-initio CCSD(T)/AV5Z calculations. In addition, an in-depth analysis of the electron charge displaced upon complex formation allows us to definitively ascertain and quantify the CT component in the halogen bond interaction, pointing at a covalent contribution and required to quantitatively reproduce the experimental cross sections.

In the Ng-X_2_ complexes (Ng = He, Ne, Ar, Kr, Xe; X = Cl, Br, I) the halogen bond formation selectively stabilizes the collinear configuration of the adduct, with X_2_ being in the *X ^1^Σ_g_* ground state. On the other hand, for the adducts in the perpendicular configuration or with X_2_ in the *B ^3^Π_u_* excited state, this effect is missing (see Figure 2, top) [17,54]. The proposed analytical model PES is composed of three additive components—(1) the two-center vdW component, represented by an Improved Lennard-Jones (ILJ) formulation [55]; (2) an exponentially decaying “three-body” term, pointing at the interaction anisotropy due to the X_2_ molecular orbitals—especially critical in the perpendicular and saddle configurations concerning the interaction of Ng with the X_2_ outermost π_g_* density; (3) a two-term exponentially decaying charge transfer component, essential for reproducing the XB interaction in the collinear ground state configuration.

Since it is well known that the ILJ function adequately reproduces the vdW interactions, any deviation of the interaction potential from the ILJ prediction suggests the occurrence of additional interaction components. Specifically, the presence of a CT component in the Ng-X_2_ complexes bond has been ascertained by applying the CD analysis (see Figure 2, bottom). For the Ng-X_2_ collinear configurations, the CD curves are distinctly positive everywhere, even for the He case, thus pointing at an electron flow in the direction from Ng to X_2_. Conversely, in the Ng-X_2_ perpendicular configurations the CD function is much smaller in magnitude and changes sign between the two interacting partners, thus pointing at a much smaller, if not negligible, CT. We can therefore unambiguously conclude that in the collinear configurations a sizable contribution to the overall interaction potential is related to a non-electrostatic CT component, that selectively concurs to the formation of the XB. By taking the value of the CD function at the isodensity boundary between the fragments as an estimate of the CT from Ng to X, we gained values ranging from 0.39 to 7.62 milli-electrons (me), with increasing values going down in the group for both the Ng and X atoms, as expected on the basis of the atomic properties.

It is worthwhile to remark that the employed CD analysis suggests the presence of a small CT component even in the helium cases, with a transferred charge of 0.39/0.47/0.55 me in He-Cl_2_/Br_2_/I_2_ complexes, to be compared with 3.33/5.59/7.62 me computed for the correspondent Xe complexes [17]. Remarkably, the ab-initio CT magnitude values decay exponentially with the inter-fragments distance, matching the exponentially decay form adopted in the semi-empirical model potential. Since in the perturbative limit (i.e., when very small CT are involved) the CT magnitude and the CT energy stabilization are expected to be roughly proportional, an estimation of the related proportional constant returns the CT energy stabilization per electron unit transferred upon complex formation. For the investigated Ng-X_2_ complexes, we gained an average proportional constant of (4.6 ± 1.0) eV/e, that matches very well with the value previously derived for the Ng-CCl_4_ complexes, (5.7 ± 1.5) eV/e.

The identified XB interaction in the collinear ground state Ng-X_2_ complexes may be related with the polar flattening, marking the strong anisotropy of the halogen electron density in X_2_ and with the σ-hole, marking a halogen electrophilic region along the bond axis on the outer X_2_ side. Both these features permit a closer approach of the interacting partners selectively in the collinear direction, so that a shift to a smaller distance of the repulsive wall is accomplished, involving an increase of the dispersion attraction and enabling the electron density transfer from Ng to X_2_. The XB formation seems to be inhibited in the perpendicular configuration or in the X_2_ (π_g_* → σ_u_*) ^3^Π _u_ excited state, where population of the σ_u_* orbital fills the σ-hole, thus strongly reducing the polar flattening along the bond direction. Accordingly, the three-body and CT terms of the analytic model potential are no longer needed, since the unique vdW term is sufficient to accurately reproduce the interaction, as also confirmed by the *ab-initio* PES calculation and by the CT absence verified by the CD analysis.

The integrated theoretical-phenomenological approach has been successfully applied to more complex halogenated systems, such as O_2_-CX_4_ (X = F and Cl) [56], thus laying the foundations for build-up reliable force fields for molecular dynamics simulations. Specifically, the measured cross section data, integrated with the ab-initio calculations, prove that O_2_-CCl_4_ complexes have a stronger interaction at both long and short range and a stronger interaction anisotropy with respect to O_2_-CH_4_ and O_2_-CF_4_ complexes. In fact, the O_2_-CCl_4_ adduct in the selected vertex (XB) configuration, with the O_2_ partner pointing directly to the halogen atom, is significantly stabilized by an additional short range component. The CD analysis returns the ∆*q* curves distinctly positive in the whole interacting O_2_-CCl_4_ system (vertex configuration), with a computed ∆*q* value at the isodensity boundary of 1.1 and 0.6 me respectively in the collinear and perpendicular orientation of the vertex isomer. This clearly points at the presence of a sizable CT component to the overall interaction potential, related to the XB formation (see Figure 3). Our investigations also suggest that the interactions of CCl_4_ with O_2_, when averaged over its relative orientations, are similar to that with the Ar atom, exhibiting a similar electronic polarizability, despite of the O-O bond anisotropy and open shell nature. Accordingly, we can safely conclude that the O_2_ molecules behaves like a spherical projectile in the interaction with CCl_4_ partner, analogously to the Ar case.

Lastly, we examined the interaction of H_2_O with CCl_4_ and CF_4_ partners, in comparison with the homologous systems involving O_2_, since the H_2_O and O_2_ show comparable isotropic molecular polarizability (1.47 and 1.60 Å^3^, respectively) [57].

The measured cross section data suggest that in the H_2_O- and O_2_-CF_4_ complexes the interaction potential depends almost exclusively on the vdW component, the CT component being almost absent, as confirmed by ab-initio calculations and CD analysis. A different picture emerges for the CCl_4_ complexes with H_2_O and O_2_, where a comparable vdW component is found but a strong CT effect seems operative in the complex with water, evidenced by a shift of the measured glory extreme positions at the higher velocity with respect to the O_2_ case. Indeed, the CD analysis points out the existence of two different stabilization effects, the former due to the occurrence of a pronounced XB, mostly for the vertex H_2_O-CCl_4_ configuration, where water approaches CCl_4_ on the oxygen side, the latter related to the presence of a hydrogen bond (HB) interaction in some specific complex geometries, where H points to Cl in orthogonal direction with respect to the C-Cl bond (see Figure 4). The CT attributed to the XB and HB interactions were almost of the same entity (6.7 and 6.4 me, respectively, at the isodensity boundary) and both contribute to stabilize in energy selected configurations of the H_2_O-CCl_4_ adduct.

### 3.2. Condensed Phase

The CD analysis has been applied also to obtain information about systems in condensed phase, solid or solution. For example, an in-depth analysis of the Cambridge Structural Database (CSD) reveals that a huge number of triiodide systems I^1^-I^2^-I^3^ exist and, more importantly, the I^1^-I^2^ and I^2^-I^3^ distances are not mutually independent. Indeed, in most cases the two distances are correlated, forming a general trend: as the former decreases, the latter increases [58]. The experimental data could be satisfactorily fitted with an equation coming from the Bond Valence (BV) theory [59,60]. Prompted by these results, we decomposed the symmetric I_3_^−^ system in I_2_ and I^−^, treating the system as an extreme case of XB. The 3D deformation maps of the electronic density showed that the iodide undergoes charge depletion and the iodine charge accumulation, with a severe polarization of the molecule (Figure 5a) [61]. A large amount of charge is displaced at the isoboundary (CT), 0.400 e, underlining the covalency of the 3c4e bond.

Then, both I^1^-I^2^ and I^2^-I^3^ distances have been modified, applying a geometrical constraint on one of them and optimizing the other. The generated systems have been decomposed in the same way, an iodide and a molecule of iodine and a CD curve was generated for each system. In all the cases, the CD curve maintained the same general shape, whereas the intensity of the curve decreases (increases) as the distance between the iodide and I^2^ increases (decreases). The value of CT goes from 0.416 to 0.256 e, also in this case without any discontinuity, underlining, as in the Rosokha’s paper [44], the similarities between covalent and non-covalent interactions.

The finding can be generalized considering all the trihalide X_3_^−^ (Figure 5b), heterotrihalide XYX^-^ and chalcogen-dihalogen systems EX_2_ (E = S, Se) [58]. In all the cases, CSD reveals that the correlation between the two bond distances is always similar and, after a normalization, all of them together with the BV equation. On the other hand, the CD study revealed that the CT is the same for all the trihalide and depends only on the (normalized) distances, not on the nature of X [61]. Not only that but all the heterotrihalides behave similarly, with a small systematic difference depending on the different polarizability of the systems.

For neutral systems involving chalcogen atoms, the CT function has again the same general shape but the extent of the polarization is much lower (0.261 instead of 0.400 e) [62]. Also in this case, the nature of E or X has a little influence on the value of CT, which depends only on the normalized distances. And, again, the CD function has always the same general shape, for short E^…^X bond distances (covalent bond) or long ones (non-covalent bond) without any discontinuity. By using the SAPT0 analysis [8,63], which is able to decompose charge transfer and polarization, it has been also shown that the CT_SAPT0_ values are very similar for ionic and neutral systems and only the polarization is different.

The CD has been used also to verify how a XB is able to influence the other interactions in which the XB acceptor is involved. For example, the hexamethylenetetramine (HMTA) has four potential XB acceptor sites, and, indeed, the solid state structure of HMTA:4 NIS (where NIS = N-Iodosuccinimide) shows that all the four nitrogen atoms are involved in XBs. On the other hand, diffusional nuclear magnetic resonance (NMR) studies demonstrated that, in solution, only two out of the four sites of HMTA can be occupied by N-Bromosuccinimide (NBS), even using a large excess of the latter ([NBS]/[HMTA] > 50) [64]. Thermodynamically, this is obviously due to a large entropic contribution which hampers the formation of large aggregates.

The reasons of this selectivity have been investigated by CD. Firstly, the XB in the 1:1 adduct has been characterized as 0.130 e, clearly from the nitrogen to the bromine (Figure 6). Interestingly, a small depletion region is visible in the nitrogen atoms not involved in the XB. Regarding the 1:2 adduct, the fragmentation scheme was [(NBS^1^)(HMTA)]^…^[NBS^2^], as only the “second” XB was under study. The resulting Δ*ρ* has been analyzed integrating along two different axes: the first one passed through the nitrogen and the bromine involved in the XB under study (NBS^2^), to evaluate its own charge transfer and the second one passed through the nitrogen and the bromine involved in the “first” XB (NBS^1^), in order to measure the effect of the “second” XB on the first one. The results nicely agreed each other, as the CT relative to the N^…^Br^2^ XB is lower than those in the 1:1 adduct (0.110 instead of 0.130 e) and the presence of an additional XB weakens the N^…^Br^1^ XB of 0.011 e. This is consistent with the small depletion regions observed in the other nitrogen atoms upon the formation of the 1:1 adduct.

Together with the analysis of Δ*E*/n (when n is the number of NBS moieties present), which already characterized these XB as anti-cooperative, the CD analysis provides the reasons for which it is anti-cooperative. Notably, in other cases the XB resulted to be cooperative [65,66,67,68].

Finally, the last example is about the interaction between a second-sphere XB and the DCD components of a M-L bond. As already mentioned in the Introduction, CD function analysis is particularly useful in the characterization of coordination bonds and the examples described above show that it can be utterly useful also for XB adducts. Therefore, the two topics have been unified in a recent contribution [37]. As a case study, [(NAC)Au(SeU)]BF_4_ (NAC = nitrogen acyclic carbene, SeU = selenourea) has been chosen—the selenium can establish XB with polarized halogen atoms [58,62,67], the NAC is an experimental probe for Au → C back-donation, due to the inverse proportionality between the latter and the C-N rotational barrier [27].

Unfortunately, this well-designed complex betrayed the authors’ trust for its intrinsic instability but the CD methodology confirmed that the system was really ideal to study the interplay between XB and back-donation. Indeed, the Au-C and Au-Se bonds have been firstly characterized in the absence of any second-sphere interaction. As the system is not symmetric, the NOCV-CD decomposition has been used. The Au-C bond revealed to be composed of 0.359 (donation) and −0.092 (back-donation) e, whereas the Au-Se bond of 0.346 and −0.037 e, respectively. The rotational barrier for the C-N bond is 19.2 kcal/mol. Upon the introduction of XB (donor: ICF_2_CF_3_), most of these parameters vary: the Au-C donation increases (0.372 e) and the back-donation decreases in absolute value (−0.080 e, see Table 1), coherently with the C-N rotational barrier that increases up to 20.0 kcal/mol. On the other hand, the Au-Se bond remains practically unchanged, probably for the compensation of different effects. The decrease of the Au → C back-donation may appear small but it should be considered that the donor/acceptor properties of the carbene are only slightly influenced by chemical modifications on the carbene structure [23,69,70], and, on the other side, activation barriers can be quite sensible to back-donation [71].

The same gold complexes, in the presence of a stronger and more structured XB donor as NIS, gave different results. In fact, in this case the iodine does not interact with the lone pair of the selenium but directly with the gold, stabilized by an additional hydrogen bond between the oxygen of the NIS and the amino proton of the SeU (Figure 7). NBO analysis of the system confirmed that the Au-I interaction is a XB between one of the lone pairs of the gold(I) and the σ*_IN_ orbital. There are some reported cases of metals as XB acceptors but not so many [72,73,74].

Also in this case, there is an impact on the DCD components, as the Au ← C donation increases from 0.359 to 0.375 e, whereas the Au → C back-donation decreases from −0.092 to −0.081 e (Table 1). Clearly, this is due to the fact that the gold is donating part of its electronic density to the iodine, it becomes more acidic and attracts more electronic density from the carbon. For the same reason, it is less prone to donate to the carbon. Similarly, the presence of the XB induces a larger Au ← Se donation and a lower Au → Se back-donation (Table 1).

These data demonstrate that second-sphere weak interactions can significantly alter the electronic properties of the metal. The implications of this deserve to be investigated in the future, for example verifying the catalytic properties of a complex in the presence and absence of a XB donor.

## 4. Conclusions

In this review, we showed how much the Charge Displacement bond analysis can be useful in the characterization and understanding of halogen bonding, both in gas phase, in solution and in the solid state. In all the cases, the coupling of CD with experimental techniques has been crucial to reach an intimate and multi-faceted comprehension of the chemical system under study.

In many cases, the combination of experimental and theoretical techniques offers a precious insight about the nature of halogen bonding, which demonstrated to have a non-negligible and tunable orbital component. This makes the halogen bond a complex interaction. Indeed, XB overcomes the “covalent”/“non-covalent” dichotomy, as it can move from one side to the other in a continuous way, depending on the nature of the two fragments and their mutual orientation.

In this framework, the CD bond analysis, which is capable to characterize and quantify the orbital component of the XB, is and will be crucial to provide a theoretical framework to interpret experimental data.

## Figures and Tables

**Figure 1 molecules-25-00300-f001:**
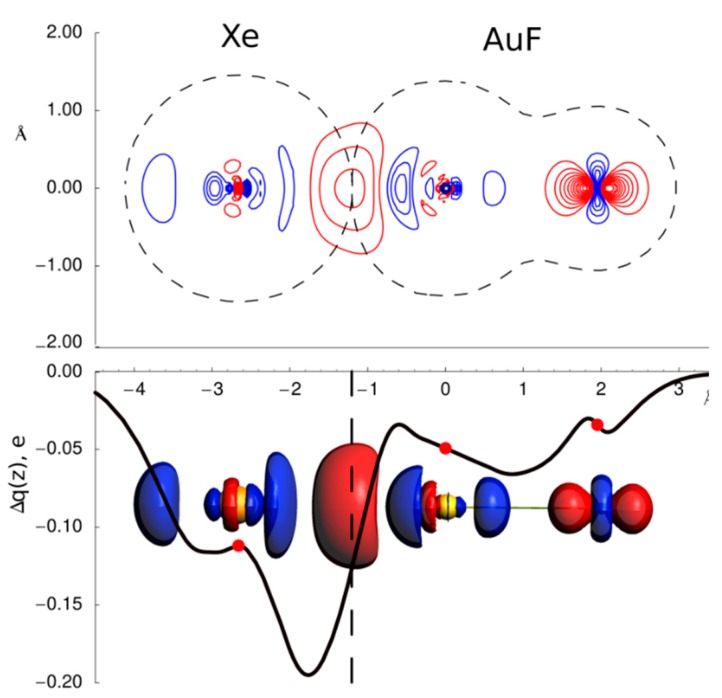
Contour plots of the electron density difference between XeAuF molecules and the Xe and AuF fragments. The positive contour levels (red) range from 0.005 to 0.2 e/bohr^3^ with a step of 0.005, while the negative ones (black) range from −0.005 to −0.2 e/bohr^3^ with a step of 0.005. In the bottom part the integrated charge displacement function Δ*q* (see text) is reported as a function of the internuclear distance. The 3D isodensity plot of the electronic charge due to the intermolecular interaction (cutoff = ± 0.005 e/bohr^−3^, with negative/positive values in blue/red).

**Figure 2 molecules-25-00300-f002:**
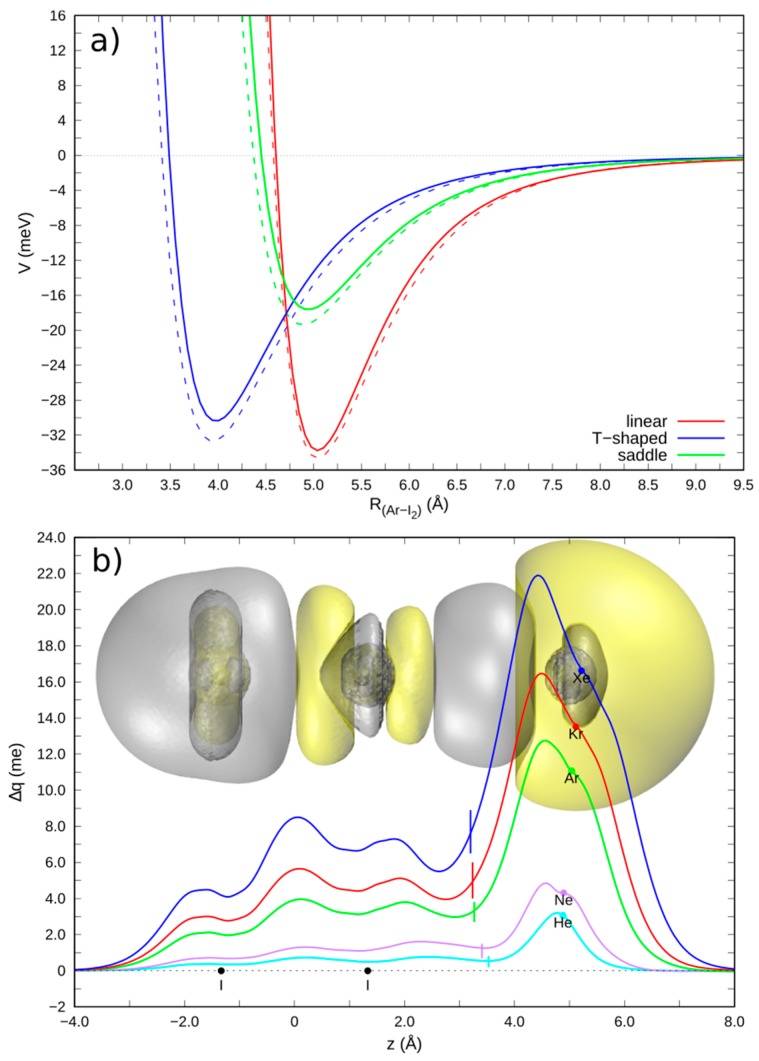
(**a**) Potential energy curves (interaction potential *V* vs. Ar-I_2_ distance *R*) for the selected ground state Ar-I_2_ complexes in the three configurations [solid lines, CCSD(T)/AV5Z level of theory; dashed lines, curves from semi-empirical model]. (**b**) CD curves of the ground state Ng-I_2_ complexes in the linear configuration - dots represent the atomic nuclei position on the z axis and vertical lines mark the isodensity boundaries. The inset shows the 3D isodensity plots of the electron density change, accompanying bond formation (∆ρ = 8∙10^−6^ me/bohr^3^, negative/positive values in yellow/silver). (adapted from Reference 17).

**Figure 3 molecules-25-00300-f003:**
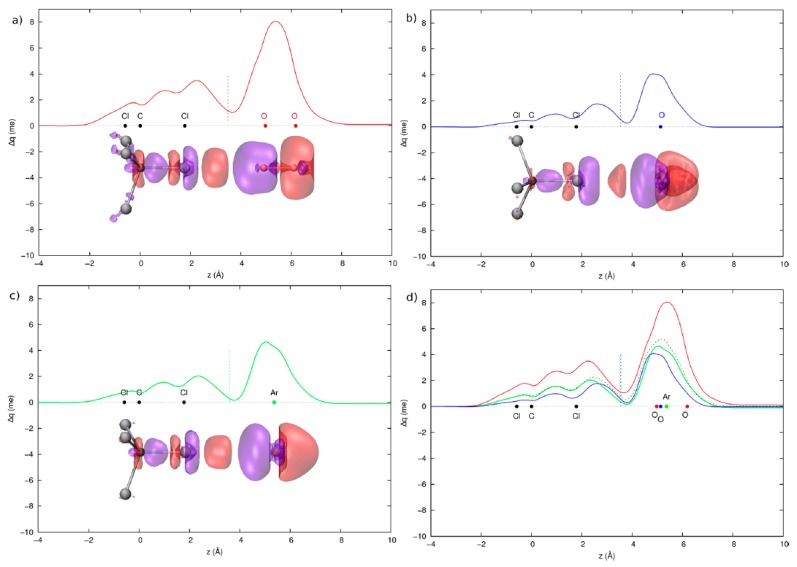
Charge displacement (CD) curves for the vertex configuration of the O_2_-CCl_4_ system in the collinear (**a**) and perpendicular (**b**) orientations of the O-O bond and for the Ar-CCl_4_ system (**c**). (**d**) CD curves for the O_2_-CCl_4_ system in the collinear (red) and perpendicular (blue) orientations and that obtained as weighted average in the ratio 1:2 (curve with dashed line), according to their degeneracy, are compared with the CD curve of the Ar-CCl_4_ system. The insets show the 3D isodensity plots of the electronic charge due to the intermolecular interaction (cutoff = ±0.05 me/bohr^3^, with negative/positive values in red/blue). (from Reference 56, Copyright © 2020, American Chemical Society, with permission.).

**Figure 4 molecules-25-00300-f004:**
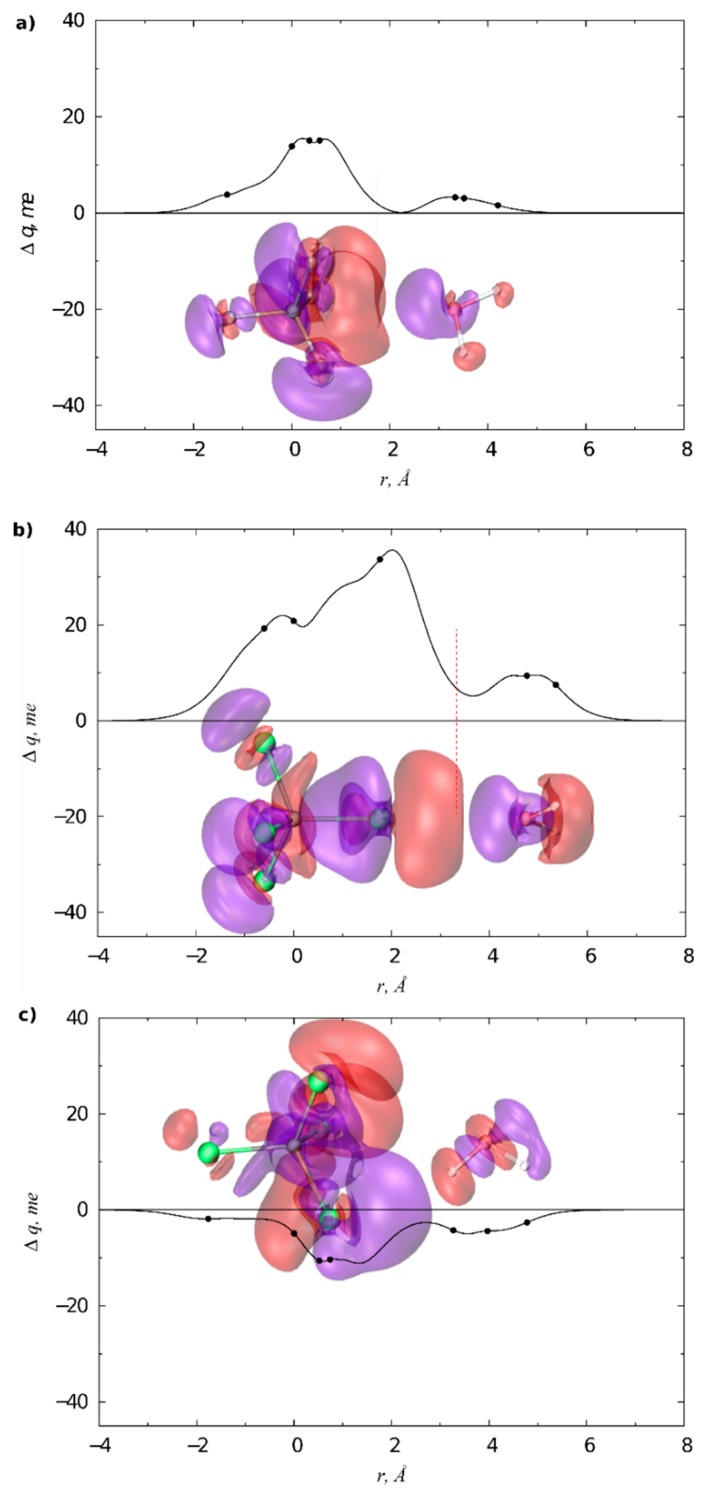
3D isodensity plots (cut-off = ±0.15 me/bohr^3^, negative/positive values in red/violet) and correspondent CD curves of the electron density change due to the intermolecular interaction for the CF_4_-H_2_O (**a**) and CCl_4_-H_2_O systems at the global minimum optimized geometries (**b**,**c**). Dots correspond to the projection of the nuclei positions on the *z* axis. The axis origin is at the tetrahedral carbon. The vertical dashed lines mark the isodensity boundary between the fragments. (Reproduced from Reference 57 with permission from the Centre National de la Recherche Scientifique (CNRS) and The Royal Society of Chemistry).

**Figure 5 molecules-25-00300-f005:**
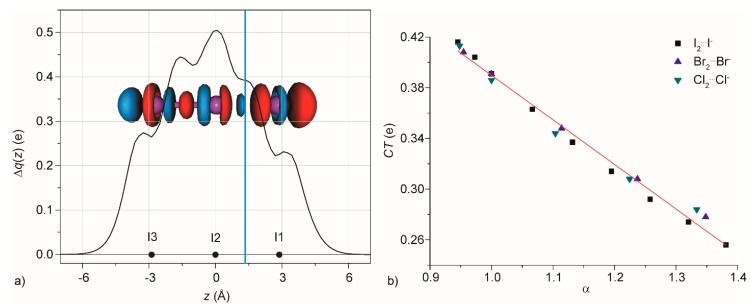
(**a**) Charge Displacement function for I_3_^−^. The black dots represent the *z* coordinate of the atoms. The light blue vertical line identifies the boundary between the two fragments. Overprint: contour map of the change of electronic density upon formation of the complex I_3_^−^ from I_2_ and I^−^. The direction of the charge flow is red → blue. Density value at the isosurfaces: ±2.5 me/bohr^3^; (**b**) Dependence of charge transfer (CT) on α (d_X1–X2_/d_X2–X3_) for trihalide systems. The black line is the best linear fit (CT = 0.73 – 0.34α, r^2^ = 0.9694). (Reproduced from Reference 61 Copyright © 2020, John Wiley and Sons, with permission.).

**Figure 6 molecules-25-00300-f006:**
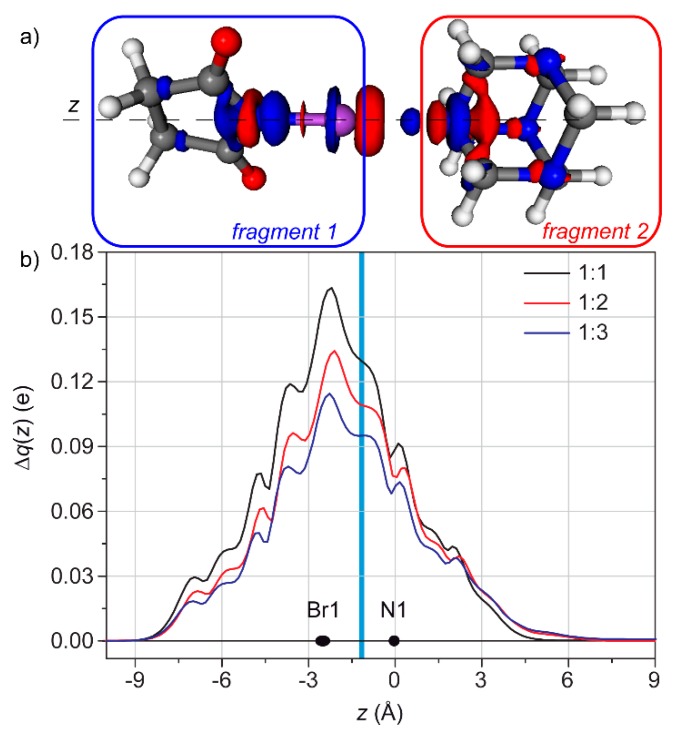
(**a**) 3D contour map of the change of electronic density upon formation of the adduct HMTA/NBS (1:1). Blue (red) isosurfaces identify regions in which the electron density increases (decreases). Density value at the isosurfaces: ± 2 me/bohr^3^. (**b**) CD functions for HMTA/NBS adducts with different stoichiometry. The black dots represent the *z* coordinate (or the range of coordinates) of the atoms. The light blue vertical band identifies the range of the inter-fragment boundaries. (Reproduced from Reference 64 with permission from The Royal Society of Chemistry).

**Figure 7 molecules-25-00300-f007:**
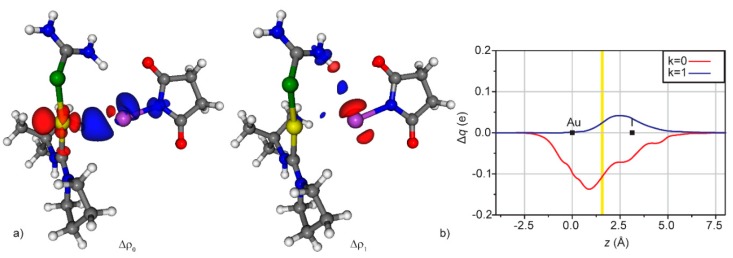
(**a**) Isodensity surfaces (±1.2 me/bohr^3^) for the deformation maps relative to the Δρ_k_ (k = 0 and 1) contributions of the [(SeU)Au(NAC)]^+…^[NIS] bond. The charge flux is red → blue; (**b**) CD functions (CD_k_, k = 0 and 1) for the [(SeU)Au(NAC)]^+…^[NIS] bond. Black dots indicate the *z* position of the atomic nuclei. A yellow vertical band indicates the boundary between the fragments. (Reproduced from Reference 37 with permission from the PCCP Owner Societies.).

**Table 1 molecules-25-00300-t001:** NOCV-CD bond analysis results for an organometallic system (charge transfers, CT_k_, in electrons).

Fragments	CT_0_	CT_1_	CT_2_	CT_3_	CT_tot,back_
[(**SeU**)Au]^+…^[**NAC**]	0.359	−0.049	−0.030	−0.013	−0.092
[**SeU**]^…^[Au(**NAC**)]^+^	0.346	0.024	−0.021	−0.016	−0.037
[(**SeU**)(ICF_2_CF_3_)Au]^+…^[**NAC**]	0.372	−0.040	−0.030	−0.010	−0.080
[(**SeU**)(ICF_2_CF_3_)]^…^[Au(**NAC**)]^+^	0.344	0.020	−0.021	−0.015	−0.036
[ICF_2_CF_3_]^…^[Au(**SeU**)(**NAC**)]^+^	−0.039	0.005	-	-	-
[(**SeU**)(**NIS**)Au]^+…^[**NAC**]	0.375	−0.044	−0.027	−0.010	−0.081
[**SeU**]^…^[Au(**NIS**)(**NAC**)]^+^	0.373	0.033	−0.018	−0.014	−0.032
[**NIS**]^…^[Au(**SeU**)(**NAC**)]^+^	−0.106	0.024	-	-	-

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
