# Peer review of "Charge Displacement Analysis—A Tool to Theoretically Characterize the Charge Transfer Contribution of Halogen Bonds"

_molecules, 2020, doi:10.3390/molecules25020300_

Round 1
Reviewer 1 Report
This is a review article of Charge Displacement Analysis (CDA) tool, with emphasis in its use for characterizing charge transfer contribution of halogen bonds. This theoretical tool was developed by Leonardo Belpassi and Co-workers [JACS, 130 (3), 1048, 2008]. In combination with experimental technique, the authors have successfully applied this methodology to study various types of chemical interactions. In this particular review, the authors present theoretical characterization of halogen bonds (XBs) in a variety of systems as well as in different environments. As the CDA method may find more extensive usage in the future, this review article is likely to have impact in both theoretical and experimental communities of chemistry. It is going to be valuable review for theoreticians and experimentalists who want to better understand halogen bonding. Only a few typos in the writing of this manuscript need to be addressed, and they are listed below. I recommend publication of this manuscript after minor revision.
Detailed suggestions:
1) [line: 56] remove “Anyway”.
2) [line: 62] dispersion…)., better use etc.
3) [line: 84] Symbol of Delta-rho is inconsistent with the equation-1
4) [line: 87] same thing with Delta_q(z)
5) [line: 92] equation (1) has a long empty space – should get rid of this.
6) [line:83 and line: 181] section 2 and section 3 have the exact same name “Methodological aspects” – need to rewrite this.
7) [line: 244] by the PES ab-initio calculation makes no sense, it should be rephrased as ab initio PES calculations
8) [line: 263] unambiguously points – should be changed to unambiguity points
9) [line: 426-27] restructure the sentence – the CD bond analysis is capable to ….of the XB, and will be ….
Author Response
We thank the referee for the detailed suggestions
"point 1) [line: 56] remove “Anyway”."
It has been removed
"2) [line: 62] dispersion…)., better use etc."
The line has been modified
"3) [line: 84] Symbol of Delta-rho is inconsistent with the equation-1"
The notations have been made consistent each other
"4) [line: 87] same thing with Delta_q(z)"
The notations have been made consistent each other
"5) [line: 92] equation (1) has a long empty space – should get rid of this."
The equation has been reformatted
"6) [line:83 and line: 181] section 2 and section 3 have the exact same name “Methodological aspects” – need to rewrite this."
Section 3 has been renamed "Applications"
"7) [line: 244] by the PES ab-initio calculation makes no sense, it should be rephrased as ab initio PES calculations"
The sentence has been rephrased
"8) [line: 263] unambiguously points – should be changed to unambiguity points"
We understand the suggestion. We prefer to use "clearly" instead of "unambiguously"
"9) [line: 426-27] restructure the sentence – the CD bond analysis is capable to ….of the XB, and will be …."
The sentence has been modified.
Reviewer 2 Report
This is a very interesting contribution to the field of so-called non-covalent interactions, the case of the halogen bond is considered here on the basis of results concerning simple complexes and with the use of the Charge Displacement Method.
Generally I do not have reservations, maybe it would be better to consider other complexes linked by the halogen bond (here mainly those containing noble gases are taken into account) but it may be treated as a first step of further analyses.
If there is any reservation from my side, it is slight one; it concerns experimental methods and methods of calculations the authors refer to in the text.
For example, ¨it has been possible to obtain pictures of single molecules [2-5]¨- it is true, AFM is given in titles of articles, but I think it should be mentioned in the text since it is one of the most promising new techniques,
or CCSD(T) is mentioned further in the text without specification of the basis set; there are other such minor shortcomings.
I think that it should be interesting for potential readers to give a richer description of techniques and methods that are mentioned in this article.
Author Response
We thank the referee for its comments
"it has been possible to obtain pictures of single molecules [2-5]¨- it is true, AFM is given in titles of articles, but I think it should be mentioned in the text since it is one of the most promising new techniques"
AFM has been added to the main text
"CCSD(T) is mentioned further in the text without specification of the basis set"
The basis set is now explicit in the main text.
"I think that it should be interesting for potential readers to give a richer description of techniques and methods that are mentioned in this article."
Indeed, it would be interesting, but the experimental methods we mention in this review are quite different each other (cross section data and diffusional NMR techniques, for example). Describing in detail all of them would make the manuscript too long and dispersive. We prefer to focus on CD and the readers interested to a specific example can read the related reference.
Reviewer 3 Report
This manuscript provides a review for the applications of charge displacement analysis scheme in halogen bonds.
I would like to recommend this article with minor revisions. Here are some concerns:
What's the definition of the complementary orbital in eq (3)? What's the relationship between the Δρ in eq (1) and Δρ' in eq (3) ? The charge displacement can be regarded as the population analysis within the geometrical space. Is it basis set dependent? "Methodological aspects" in Row 181 should be "Application aspects"Author Response
"What's the definition of the complementary orbital in eq (3)?"
The orbitals are generated by using the "valence operator" reported in equation (2) (see equation (3)). Indeed, "complementary" should be intended in a broad sense. We used the quotation marks in the revised version.
"What's the relationship between the Δρ in eq (1) and Δρ' in eq (3) ?"
The two quantities are computed in a totally different ways, but both of them refer to the density difference of the adduct woth respect to the components. For this they have similar notations (Δρ) , but the presence of the apostrophe makes the two clearly recognizable. A sentence has been added to the main text to calrify this.
"The charge displacement can be regarded as the population analysis within the geometrical space. Is it basis set dependent?"
Actually, CD demonstrated to be quite stable with respect to computational details. A sentence and a reference have been added.
""Methodological aspects" in Row 181 should be "Application aspects" "
Section 3 has been renamed "Applications"